# Real-World Implementation of PRISMA-7 and Clinical Frailty Scale for Frailty Identification and Integrated Care Activation: A Cross-Sectional Study in Northern Italian Primary Practice

**DOI:** 10.3390/jcm14103431

**Published:** 2025-05-14

**Authors:** Angelika Mahlknecht, Christian J. Wiedermann, Verena Barbieri, Dietmar Ausserhofer, Adolf Engl, Giuliano Piccoliori

**Affiliations:** Institute of General Practice and Public Health, College of Health Care Professions, Lorenz Boehler-Street 13, 39100 Bolzano, Italygiuliano.piccoliori@am-mg.claudiana.bz.it (G.P.)

**Keywords:** frailty screening, PRISMA-7, Clinical Frailty Scale, primary care, personalised medicine, integrated domiciliary care

## Abstract

**Background/Objectives**: Frailty screening is crucial for identifying vulnerable older adults who may benefit from interventions. However, the implementation of screening in primary care and integration into personalised care pathways remains limited. This study examined the feasibility of a two-step frailty screening approach combining PRISMA-7 and the Clinical Frailty Scale (CFS). The study assessed PRISMA-7 cut-offs’ impact on frailty classification, CFS agreement, and activation of integrated domiciliary care. **Methods**: This cross-sectional study was conducted in Northern Italy. General practitioners screened patients aged ≥75 years using the PRISMA-7 tool; if the result was positive (score ≥ 3), the Clinical Frailty Scale (CFS) was subsequently applied. Descriptive statistics, group comparisons, correlation analyses, and logistic regression models were employed to evaluate the predictors of frailty and activation of integrated domiciliary care. Comparisons were made for PRISMA-7 cut-off values ≥3 and ≥4. **Results**: Among the 18,658 patients evaluated using PRISMA-7, 46.0% were identified as frail with a threshold of ≥3 and 28.8% with ≥4. In a subset of 7970 patients assessed using both PRISMA-7 and the Clinical Frailty Scale (CFS), CFS confirmed frailty (score ≥ 5) in 48.3% of the patients at a PRISMA-7 cut-off of three and 68.2% at a cut-off of four. The female sex predicted frailty by CFS, whereas the male sex was correlated with frailty at the PRISMA-7 cut-off of three. Rural location was correlated with frailty by PRISMA-7 but showed an inverse relationship with frailty by CFS. Integrated domiciliary care began in 14.2% of the patients meeting the clinical criteria, with a higher frequency in rural areas. Concordance between PRISMA-7 and CFS increased with patient age, and at a cut-off of four. **Conclusions**: Two-step frailty screening using PRISMA-7 and CFS is viable for primary care. Using a PRISMA-7 cut-off score of ≥4 may reduce frailty overestimation, enhance congruence with clinical assessments, and reduce sex-related bias. These findings support incorporating structured screening into personalised care planning and refining frailty tools to improve equity and effectiveness.

## 1. Introduction

The increasing proportion of older adults necessitates that primary healthcare systems prioritise the early and effective identification and management of frailty. Frailty, characterised as a multidimensional syndrome of diminishing physiological reserves, is linked to increased risks of adverse outcomes, including falls, disability, hospitalisation, and mortality [1,2,3]. The true clinical significance of identifying frailty extends beyond risk stratification, as it also holds the potential to inform personalised care strategies that address individual patient vulnerabilities and promote autonomy [4,5].

Recent health policy reforms, such as Italy’s Ministerial Decree 77/2022, have mandated the incorporation of frailty screening into routine general practice [6]. This development reflects the growing recognition of frailty as a fundamental aspect of person-centred care, enabling clinicians to tailor monitoring, prevention, and resource allocation to meet individual needs. However, the implementation of such measures in everyday primary care settings remains insufficiently explored, particularly concerning the translation of screening results into actionable interventions [7,8].

Among the various frailty assessment tools available, PRISMA-7 and the Clinical Frailty Scale (CFS) are frequently recommended because of their user-friendliness and clinical applicability [9,10]. PRISMA-7 is a succinct self-reported questionnaire that evaluates mobility, support requirements, and social determinants, initially developed to identify individuals with disabilities at risk of functional decline [11]. Conversely, the CFS relies on physician judgement to assess overall functioning and frailty on a 9-point scale, and is well established in clinical environments [12]. A recent meta-analysis confirmed the diagnostic validity of PRISMA-7, with pooled sensitivity and specificity estimates of 73% and 86%, respectively [13]. Comparative studies have indicated that CFS provides higher specificity but lower sensitivity than PRISMA-7, supporting its complementary use in two-step screening models [10,14,15]. This study is built on this framework. In real-world primary care setting in Northern Italy, we implemented a structured two-step frailty screening strategy using PRISMA-7 and CFS among community-dwelling adults aged 75 years and older. As a key innovation, we linked the screening results to integrated domiciliary care (IDC) activation, an established yet underutilised care pathway in Italy that facilitates multidisciplinary home-based interventions for frail patients [16].

The aim of this study was to evaluate the real-world application of PRISMA-7 and the CFS in primary care. Specifically, it aimed to compare the two PRISMA-7 thresholds (cut-off ≥3 vs. ≥4) to identify the demographic predictors of frailty classification, and to examine the profile of individuals for whom IDC was activated.

By connecting frailty identification with individualised care activation, this study offers novel insights into the operationalisation of frailty screening as a tool for personalised medicine in general practice [4,5,8].

## 2. Materials and Methods

### 2.1. Study Design, Setting, and Recruitment of Participants

This cross-sectional study was conducted between 1 October 2023 and 31 March 2024 in South Tyrol, Italy, and was scientifically designed by the Institute of General Practice and Public Health of Bolzano. The study was integrated into routine primary care by the local National Health Service as a project with voluntary adhesion of the general practitioners (GPs) and performance-based remuneration. All 295 practising GPs were eligible and invited to participate via email.

The participating GPs were instructed during online meetings and received written guidance on study procedures. They autonomously included eligible patients (community-dwelling adults aged ≥75 years) who provided consent to participate. The GPs were instructed not only to invite patients who already displayed clinical signs of frailty but also to all assisted patients aged ≥75 years. Residents of nursing homes and individuals with insufficient German/Italian language skills were also excluded.

### 2.2. Frailty Screening/Assessment and Collection of Data

The study procedure consisted of the following three steps:Step 1. The GPs were asked to identify all their community-dwelling patients aged ≥75 years from their electronic medical records (EMRs) and to invite them to a face-to-face visit, where the patients were informed about the study procedures and signed informed consent if they agreed to participate. The participating patients autonomously completed the PRISMA-7 questionnaire.Step 2. All the patients with a resulting PRISMA-7 score ≥3 were additionally assessed by the GPs using the Clinical Frailty Scale (CFS) during the same or additional face-to-face contact, and by consulting the GPs’ EMR. The available German [17,18] and Italian versions [19,20] of PRISMA-7 and CFS were used, as these are the two main languages spoken in the investigated (bilingual) region. Owing to the need for feasibility in daily practice, the second-step CFS assessment was limited to patients with PRISMA-7 ≥ 3. Therefore, measures of diagnostic accuracy (e.g., sensitivity, specificity, and ROC curves) could not be computed using this study design.Step 3. The patients with a CFS score ≥ 5 were potentially eligible for IDC activation if not already receiving structured home-based services. The concerned cases were evaluated by GPs in cooperation with the patients themselves, their relatives/caregivers, and the nurses of the healthcare district. Only newly activated IDC cases were recorded in this study. In uncertain cases, the research team contacted GPs for clarification.

All the data (overall PRISMA-7 score and CFS score per patient, newly activated IDC, and demographic characteristics) were pseudonymously recorded by the GPs in an Excel file and returned to the research team for analysis. The GPs had the time to submit their data files until 30 June 2024. Because the study was embedded into routine care, and the collected parameters could not be automatically extracted from the EMRs, a limited set of variables was assessed.

### 2.3. Statistical Analysis

The data were analysed using IBM SPSS Statistics 27.0. Descriptive statistics are presented as absolute/relative frequencies, medians, and interquartile ranges (IQRs).

The CFS results were categorised as ‘no frailty’ (score 4), ‘mild frailty’ (score 5), ‘moderate frailty’ (score 6), and ‘severe frailty’ (score 7–9) [12] and were additionally dichotomised into ‘not frail’ (score 1–4) and ‘frail’ (score 5–9). For PRISMA-7, a binary categorisation was used (‘not frail’: score 0–2; ‘frail’: score ≥ 3) [13] according to the literature which mostly applies a cut-off of 3 [15,21,22,23]. The PRISMA-7 user guide also mentions the possibility of using a cut-off of 4 [24], thus increasing specificity and lowering sensitivity [13]. We, therefore, additionally conducted the PRISMA-7 analyses with a cut-off of 4 (dichotomization: ‘not frail’: score 0–3; ‘frail’: score ≥ 4).

In cases of missing responses, the concerned patient or GP was excluded from the analysis of the respective variables. For the analyses of CFS and IDC, all the patients were treated according to the study protocol (CFS score considered in case of PRISMA-7 score ≥ 3, IDC-activation considered in case of PRISMA-7 score ≥ 3 and CFS score ≥ 5). For the combined analyses of PRISMA-7 and CFS, the following cases were excluded: completely missing PRISMA-7 score and CFS score, missing PRISMA-7 scores, missing CFS scores, and PRISMA-negative and CFS-positive cases (Appendix A).

The rurality of the participating GP offices was defined according to the local official classification which identifies eight towns as urban and all the other villages as rural areas. Owing to data protection constraints, the patient addresses were not collected. Therefore, rurality was approximated on the basis of the officially classified location of the GP office. Although not individually verified, this represents a valid proxy in our context, as South Tyrolean GPs typically assist patients from their surrounding geographic areas.

Chi-square tests, Fisher’s exact test, Mann–Whitney U-tests and Kendall-Tau-b correlations (as appropriate) were used for univariate comparisons and correlation analyses. Kaplan–Meier estimates and accumulated Kaplan–Meier plots were used to assess the estimated prevalence of frailty over the years and to estimate the mean age at survival until frailty occurred for both sexes. Cox regression analysis was used to assess significant differences between the sexes. Due to the stepwise study design, where CFS was only applied in PRISMA-7-positive patients, diagnostic accuracy measures, such as sensitivity, specificity, and ROC analyses, could not be conducted.

Moreover, we used logistic regression models to investigate the potential predictors of frailty and IDC activation (frailty: PRISMA-7 cut-off 3, PRISMA-7 cut-off 4, CFS, and comparison of three patient groups according to their combined PRISMA-7 and CFS frailty status). Variables that were significant in univariate analyses were included as independent variables (potential predictors). We explored the correlations between the dependent variables and possible predictors as well as associations between the independent variables using Kendall-Tau-b correlations; the independent variables were excluded from the logistic regression model when they were highly correlated with another predictor (τ > 0.3). For all the significant independent variables, the odds ratios (ORs) and corresponding 95% confidence intervals (CIs) were presented. The model fit was assessed using Nagelkerke’s R^2^. Due to the large sample size, the significance level was set at *p* < 0.001 for all the tests.

## 3. Results

### 3.1. Study Population and Screening Completion

Of the 295 GPs invited to participate, 142 (48.1%) participated in this study. The median age of the participating GPs was 47 years (IQR, 39–58 years), and 50.7% were female. GPs working in urban areas (52.8%) were more represented than those working in rural areas (47.2%). On average, the participating GPs assisted approximately 1600 inhabitants, including a median of 190 persons aged ≥75 years. The participating GPs were significantly younger than their non-participating colleagues (median age 58 years, IQR 45.5–64, *p* < 0.001), and more often worked in urban locations (52.8% vs. 36.6%, *p* = 0.007). No statistically significant differences were observed in the GP sex (Appendix A).

A total of 19,501 community-dwelling older adults were enrolled, of whom 60.0% were female, with a median age of 81 years (IQR 78–85). This represents 67.3% of all the inhabitants aged ≥75 years (*n* = 27,727) under the care of the participating GPs and 32.6% of the entire ≥75-year-old population in South Tyrol (*n* = 57,190) [25].

Among the participants, 18,658 (95.7%) completed the PRISMA-7 questionnaire and were included in the frailty screening analysis. Missing PRISMA-7 and/or CFS scores occurred in 7.5% of the cases. Specifically, 829 participants had missing scores (4.3%), 14 had only PRISMA-7 missing (0.1%), and 612 were PRISMA-positive, but had no documented CFS (3.1%) (Appendix A). Based on the analysis protocol, 18,004 participants were included in analyses using a PRISMA-7 cut-off of three and 17,670 in analyses using a cut-off of four. Table 1 summarises the characteristics of the participating GPs and individuals.

### 3.2. Frailty Classification and Agreement Between Tools

Frailty according to PRISMA-7 with a cut-off ≥3 was detected in 8582 patients (46.0% of all the screened patients). Among these, 7970 patients were further assessed using the CFS, with 3852 (48.3%) classified as frail (CFS score ≥ 5). When using the alternative PRISMA-7 cut-off ≥4, the frailty prevalence decreased to 5372 patients (28.8%), while agreement with the CFS improved markedly; 68.2% of the PRISMA-7-positive individuals were also rated as frail by CFS.

The median PRISMA-7 score was 4.0 (IQR 3.0–5.0) in the frail group, with a cut-off value of three, and 5.0 (IQR 4.0–6.0), a cut-off of four. The median CFS score of frail individuals was 6.0 (IQR 5.0–7.0). For both instruments, the mildest frailty categories were most frequently observed: PRISMA-7 score, three (17.2%); PRISMA-7 score, four (10.5%); and CFS score, five (19.1%). The proportion of individuals rated in the most severe category was 2.2% for PRISMA-7 (score seven) and 0.5% for the CFS (score nine). The PRISMA-7 and CFS scores were significantly correlated (τ = 0.603, *p* < 0.001; Kendall-Tau-b). Table 2 summarises the frailty prevalence and score distributions for both tools.

The patients were classified into three frailty groups based on the combined PRISMA-7 and CFS results: not frail according to PRISMA-7, frail according to PRISMA-7 but not frail according to CFS, and frail according to both PRISMA-7 and CFS.

At a PRISMA-7 cut-off of three, 55.7% of the patients were classified as not frail, 22.9% as frail by PRISMA-7 only, and 21.4% as frail by both tools. When using a cut-off of four, the proportion of non-frail individuals increased to 71.4%, and disagreement (PRISMA-frail, CFS-not-frail) decreased markedly to 9.1%. The group classified as frail using both tools remained relatively stable (19.5%) (Table 3).

### 3.3. Frailty Classification Patterns and Determinants

The frailty prevalence varied according to the assessment tool and threshold. Using a PRISMA-7 cut-off of ≥3, 46.0% of the patients were classified as frail. This decreased to 28.8% with cut-off ≥4. Among the patients with PRISMA-7 ≥ 3 who also completed a CFS assessment, 48.3% were frail (CFS ≥ 5), showing moderate agreement between the tools. Concordance increased with age from 39.4% in patients aged 75–84 years to 56.2% in those aged ≥85 years.

#### 3.3.1. Age-Related Patterns

Figure 1 (top row) illustrates the age-associated shift in the score distributions. PRISMA-7 showed a stronger correlation with age (τ = 0.390) than CFS (τ = 0.192), although CFS was only applied to the PRISMA-positive cases.

Kaplan–Meier curves illustrate the mean age at frailty onset by sex and tool. For a PRISMA-7 cut-off score of three, women became frail later than men (mean age 87.6 vs. 85.7 years, *p* < 0.001) (Figure 2). This difference was smaller with a cut-off of four and absent for CFS. Frailty prevalence increased with age across all the tools and thresholds. The year-by-year evolution of frailty status for both PRISMA-7 cut-offs illustrated this trend in more detail (Appendix A).

#### 3.3.2. Sex-Related Differences

Figure 1 (middle row) shows that women tended toward higher CFS scores, whereas men peaked at a PRISMA-7 score of three. PRISMA-7 cut-off three classified more men than women as frail (51.5% vs. 41.9%), whereas CFS identified more women (56.8% vs. 38.9%). This discordance declined with PRISMA a cut-off of four.

Logistic regression confirmed that female sex was a predictor of frailty per CFS (OR 1.88, *p* < 0.001), but male sex was a predictor for PRISMA-7 frailty at a cut-off of three (OR for female sex 0.54, *p* < 0.001) (Table 4). Sex-based score disagreement was the most frequent among younger patients and was reduced using a cut-off of four. The full stratifications are provided in Appendix A.

#### 3.3.3. Geographic Trends

Frailty was more frequent in rural areas by PRISMA-7 cut-off three (48.0% vs. 44.7%, *p* < 0.001), but not with a cut-off of four. In contrast, CFS indicated higher frailty in urban areas (50.7% vs. 45.1%, *p* < 0.001). As shown in Figure 1 (bottom row), the PRISMA scores peaked at 3–4 in both settings, while the CFS scores showed a broader spread in urban patients.

Regression models showed rural origin as a predictor of PRISMA-7 frailty at a cut-off of three (OR 1.19), but a negative predictor for CFS-based frailty (OR 0.83) (Table 4). The extended models are listed in Appendix A.

### 3.4. Clinical Implications: Integrated Care and Variability

#### 3.4.1. Activation of Integrated Care

Among the patients who met the criteria for care activation (PRISMA-7 score ≥ 3 and CFS score ≥ 5), IDC was initiated in 526 individuals, corresponding to 14.2% of the eligible cases. The median number of IDC activations per GP was three (IQR, 1–5), ranging from 0 to 18. Three GPs with implausibly high IDC counts (*n* = 151 patients) were excluded from this analysis after failing to respond to the data validation requests (Table 5, panel A).

IDC activation occurred more frequently in rural areas (17.1%) than in urban areas (12.4%; *p* < 0.001). No significant differences were observed in patient sex (*p* = 0.410), whereas the older patients (≥85 years) showed slightly higher IDC activation rates than those aged 75–84 years (15.3% vs. 12.4%, *p* = 0.013) (Table 5, panel B).

In the multivariable logistic regression models (Table 5, panel C), rural GP office location and frailty severity were both independently associated with an increased likelihood of IDC activation. Owing to the high collinearity between the PRISMA-7 and CFS scores (τ = 0.603, *p* < 0.001), two separate models were computed. The CFS-based model demonstrated better explanatory power (Nagelkerke’s R^2^ = 0.149) than the PRISMA-7 model (R^2^ = 0.076). In both models, younger age at GP was significantly associated with IDC activation, whereas GP sex, patient age, and sex were not significant predictors.

Notably, the CFS score was a stronger predictor of IDC activation (OR 2.29, 95% CI 2.07–2.54; *p* < 0.001) than the PRISMA-7 score (OR 1.65, 95% CI, 1.50–1.81; *p* < 0.001), underlining the relevance of physician-rated functional assessments in care decisions.

Despite the systematic frailty screening process, the relatively low IDC activation rate (14.2%) suggests possible barriers to translating frailty identification into coordinated care action.

#### 3.4.2. Variability in Frailty Classification

We also explored the variability in frailty classification at the level of individual GPs. The median proportion of patients classified as frail per GP was 44% using a PRISMA-7 cut-off of three (IQR 40–52%) and 29% using a cut-off of four (IQR 24–34%). According to the CFS, the median GP-level frailty rate among PRISMA-positive patients was 46% (IQR, 35–60%). A few outlier GPs classified nearly all the assessed patients as frail, with two GPs recording a 100% CFS-based frailty rate. These findings suggest heterogeneity in tool application and clinical interpretation. The full distribution details are provided in the Appendix A.

To explore the variability in implementation, we analysed the distribution of frail patients per GP. According to the PRISMA-7 cut-off of three, the median proportion of frail individuals per GP was 44% (IQR, 40–52%), which decreased to 29% (IQR, 24–34%) with a cut-off of four. According to the CFS, the median proportion of frail individuals among the PRISMA-7-positive patients was 46% (IQR 35–60%). Some individual GPs exhibited unusually high classification rates; notably, two GPs rated 100% of their assessed PRISMA-7-positive patients as frail according to the CFS.

These findings highlight the variability in the clinical application of frailty tools, possibly reflecting differences in interpretation, experience, or documentation practices. The full distributions are presented in Appendix A.

## 4. Discussion

This large-scale cross-sectional study provides a detailed overview of frailty in older adults in northern Italy using both the PRISMA-7 screening tool and the Clinical Frailty Scale (CFS) in a two-step approach. To our knowledge, this is the first study in Italy to comparatively analyse the two PRISMA-7 cut-offs (≥3 and ≥4) against CFS in routine primary care. The findings highlight key differences in frailty prevalence, classification concordance, and influence of sociodemographic factors on screening outcomes.

Using the commonly applied PRISMA-7 cut-off of ≥3, nearly half of the screened population (46.0%) was classified as frail. This proportion dropped to 28.8% with the higher cut-off ≥4, suggesting a substantial reduction in potential overclassification. Among patients assessed with both PRISMA-7 and CFS, only 48.3% were concordantly identified as frail using a cut-off of three, whereas concordance rose to 68.2% with a cut-off of four. These results support the interpretation that a higher threshold improves alignment with clinical judgement, as operationalised through CFS [13].

Despite this improvement in concordance, the subgroup of patients who were classified as frail by PRISMA-7 but not by CFS (“PRISMA+/CFS−”) remained sizable at a cut-off of three and was markedly reduced with a cut-off of four. In contrast, the group “frail by both tools” remained stable across cut-offs, indicating that those with more manifest frailty were robustly detected regardless of the threshold. These findings suggest that a PRISMA-7 cut-off of four may provide better specificity for identifying individuals with clinically relevant frailty, particularly in settings where second-step assessments or care pathways depend on screening results [13,15].

Frailty classification was strongly age-dependent across both tools, with prevalence increasing steadily from 75 to 95 years and above, as supported in prior studies [26,27]. Sex-specific patterns differed between tools: PRISMA-7 identified more men as frail (especially at a cut-off of three), while CFS more frequently classified women as frail [28,29,30,31]. These contrasting patterns are consistent with the previous literature and underscore the importance of sex-sensitive frailty assessment.

Rural patients were more frequently classified as frail by PRISMA-7, whereas urban patients more often met the CFS criteria, reflecting potentially different social and environmental risk profiles [32,33,34]. These opposing trends may be partially explained by PRISMA-7 items referencing functional or social support (e.g., item 6) [9], which may behave differently across rural and urban contexts.

Finally, frailty was significantly associated with the activation of IDC, particularly among the patients with concordant PRISMA-7 and CFS frailty classifications. IDC was more frequently initiated in rural areas and by younger general practitioners, possibly reflecting differing organisational contexts and professional attitudes [35,36,37]. These results highlight the relevance of frailty screening in promoting intensified care and the potential for targeted implementation strategies to improve equity and uptake.

### 4.1. Comparison with Other Studies

The frailty prevalence observed in this study is consistent with prior investigations in both Italian and international settings, although methodological differences complicate direct comparisons. A previous analysis in Italy using an electronic frailty index derived from general practitioners’ electronic medical records (EMRs) reported lower but increasing rates of moderate-to-severe frailty over the past decade (from 4.4% in 2011 to 8.1% in 2021) [16]. In a northern Italian region (Friuli-Venezia Giulia), population-level screening using PRISMA-7 in adults aged ≥75 years revealed a prevalence of 30% [19]. These findings fall within the range observed in the present study using a cut-off of four (28.8%).

PRISMA-7 prevalence estimates from primary care studies generally range between 25 and 35% using a cut-off of three and around 12% using a cut-off of four [23,38], indicating that the lower threshold may overclassify individuals, especially in younger age strata. Our findings corroborate this pattern, particularly among patients aged 75–84 years, where the prevalence dropped substantially with a cut-off of four. This is in line with previous validation studies that have proposed a higher specificity at a cut-off of four, especially for healthier or younger older adults [39,40,41].

Although direct comparisons with CFS are limited due to the tool’s selective application in our cohort, the observed CFS frailty prevalence (48.3% among PRISMA-7-positive patients) was higher than the 23–25% reported in the two Italian primary care studies that applied CFS to unselected populations aged ≥65 years [42,43]. However, these differences likely reflect differences in the age composition and the two-step design used in our study. An emergency department study comparing PRISMA-7 and CFS reported an even higher prevalence (72% and 27%, respectively), likely because of clinical acuity and case selection [22].

International data show wide variability depending on the population, instrument, and setting. A systematic review reported frailty prevalence rates ranging from 3.8% to 70.6% across older populations using different tools [44], while Italian cohort studies found rates between 4% and 45% depending on definitions [30,32,45,46,47]. In England, where frailty screening has been implemented nationally in primary care since 2017, using a variety of clinician-judged tools, the detected prevalence was 35.4% among adults aged ≥65 years [48]. An Australian study using the FRAIL scale found that 33% of the patients aged ≥75 years were frail and 47% were pre-frail in general practice settings [49], confirming the high prevalence of at-risk individuals in community care.

Taken together, our findings contribute to the international literature by reinforcing that instrument choice and cut-off selection have a major impact on the measured frailty prevalence. They also confirm that PRISMA-7 a cut-off of four may strike a better balance between identifying clinically meaningful frailty and minimising misclassification, as previously suggested in multiple international validations [39,40,41,50].

### 4.2. Equity and Implementation Considerations

The results of this study highlight equity considerations in frailty screening, particularly in terms of sex and geography. Although frailty is generally more prevalent in women because of their longer life expectancy and greater risk of multimorbidity and disability [16,28,30,31,51,52], we observed an inverse pattern with a PRISMA-7 cut-off of three, which classified more men than women as frail. This discrepancy was reduced, but not eliminated, when a cut-off value of four. In contrast, CFS classified more women as frail, which is consistent with previous studies [31,51].

These findings support earlier concerns that item 2 of PRISMA-7 (assigning one point for male sex) may introduce structural bias by inflating frailty scores in men [29,39,52]. While this item may have been intended to flag higher risk profiles among men, its effect appears to be a distortion of the sex-equitable frailty classification. Several authors have proposed removing item 2 to form a “PRISMA-6” version with improved internal consistency and reduced sex-related misclassification [52]. In our study, a PRISMA-7 cut-off of four was more effective in reducing the male overclassification effect but did not fully restore balance. These results further support the recommendation of adopting a modified PRISMA-6 or adjusting the scoring threshold based on sex-stratified performance metrics.

Geographic disparities have emerged in recent years. Patients from rural areas were more frequently identified as frail by PRISMA-7, particularly at a cut-off of three, whereas CFS indicated higher frailty rates in urban areas. This contrast may be partly explained by social context: item 6 of PRISMA-7 addresses social support (“can you count on someone if you need help?”) which may yield higher frailty scores in urban settings, where social isolation is more prevalent [32,33,34,44]. However, the intent of the item is controversial: while a “no” answer may reflect autonomy and capability, it may also indicate vulnerability due to a lack of available support [24]. Prior studies from Brazil, China, and Greece have suggested that removing item 6 improves scale performance and construct validity [9,39,40].

Variations in frailty classification across GPs were notable. While PRISMA-7 showed moderate variability across providers, CFS assessments were more dispersed, with some GPs classifying nearly all the patients as frail, and others showing greater discrimination. This reflects both the semi-structured nature of the CFS and the influence of clinician judgement, which may differ by experience, comfort with geriatric concepts, or familiarity with the tool. Although PRISMA-7 has demonstrated strong inter-rater reliability in prior studies [39], the observed GP-level heterogeneity in frailty identification supports the need for consistent training, calibration, and possibly audit feedback strategies in scaling up screening programmes.

Taken together, these findings suggest that frailty screening tools are not neutral. Their structure, scoring, and implementation context influence who is identified as frail and who gains access to further care. To ensure equitable detection and management, screening tools should be regularly validated across sociodemographic groups and their item composition should be revisited where structural bias is likely.

### 4.3. Integrated Domiciliary Care (IDC) Insights

The implementation of the IDC in this study setting provides insights into how frailty screening may translate into clinical action. Among the patients classified as frail by both PRISMA-7 (cut-off ≥ 3) and CFS, IDC was activated in 14.2% of the cases. While this figure may seem modest, it reflects real-world uptake in an existing regional model where integrated care is available, but not uniformly triggered by frailty.

Frailty status was the strongest predictor of IDC activation in both logistic regression models. CFS showed a higher predictive value (OR 2.29, 95% CI 2.07–2.54) than PRISMA-7 (OR 1.65, 95% CI 1.50–1.81), likely reflecting its closer alignment with clinician judgement of severity and functional decline. These findings reinforce the utility of two-step frailty models in clinical decision making, in which a brief screener (e.g., PRISMA-7) is followed by a more nuanced clinical assessment (e.g., CFS) to guide intervention planning [4,53].

Interestingly, IDC was activated more often in rural areas than in urban settings (17.1% vs. 12.4%, *p* < 0.001). This disparity may reflect differences in care organisation; rural regions may benefit from closer interprofessional networks, stronger informal support systems, and simplified care coordination [33,34]. In contrast, urban areas—despite often having more resources—may face bureaucratic fragmentation, workforce shortages, and communication barriers that hinder care integration. These structural challenges warrant targeted health policy interventions to ensure equitable IDC access across geography.

GP-related factors also influence IDC uptake. Younger GPs were more likely to activate IDC, which may reflect generational differences in attitudes toward team-based care or greater receptiveness to integrated models. Alternatively, less-experienced GPs may feel a stronger need to involve multidisciplinary teams for reassurance and shared management. In contrast, the sex of the GP was not associated with IDC activation.

While the IDC model in this study region ensures continuity of medical care, it does not systematically include all the components of a comprehensive geriatric assessment, such as structured medication reviews, nutritional optimisation, or physical activity promotion [5,21]. This may limit its potential impact on the reversal and mitigation of frailty. Nonetheless, its widespread availability and compatibility with routine practice make it a pragmatic choice for this study. Future enhancements could include more structured assessment modules or allied health involvement to strengthen their preventive and rehabilitative capacity.

Despite these strengths, the overall activation rate of IDC remained low in proportion to eligible patients, suggesting that additional barriers, such as unclear referral pathways, limited nursing capacity, or variation in GP engagement, may have contributed to underutilisation. These findings reflect a broader gap in the Italian primary care system, where frailty is recognised but not yet systematically addressed by structured care pathways [16].

A planned follow-up study will assess the impact of IDC activation on clinical outcomes including hospitalisation and mortality. These data will be essential to quantify the added value of frailty screening in directing patients toward intensified person-centred care.

### 4.4. Strengths and Limitations

This study had several strengths. First, it represents one of the largest population-level frailty screening efforts conducted in Italian primary care. The prospective design and structured two-step assessment using both PRISMA-7 and CFS enhanced the reliability and depth of the findings. By integrating data from over 140 GPs and nearly 20,000 older adults, this study reflects the real-world diversity in practice settings and patient populations. The use of two validated tools also supports the growing recommendation for stepwise screening approaches to improve detection accuracy in primary care [15].

Importantly, the study leveraged an existing care pathway, IDC, which enabled the investigation of how frailty classification may influence care activation. The tools were implemented with minimal disruption to routine practice and without additional resources or equipment, demonstrating the feasibility of scalable frailty screening.

However, this study had some limitations. The participation rate among the invited GPs was moderate (48.1%), and participating GPs differed from non-participants in terms of age and practice location. This raises the possibility of a selection bias and limits the generalisability of the findings beyond the study region. Furthermore, the participating GPs may have been more motivated or more familiar with structured assessments, which could have influenced both the quality and consistency of the frailty classification.

CFS was only applied in patients with a PRISMA-7 score ≥ 3, precluding a full comparative analysis of test accuracy (for example, ROC curve analysis or sensitivity/specificity estimates). Additionally, only summary scores for PRISMA-7 were collected; item-level data were not available, which limits a more detailed psychometric analysis or exploration of specific sources of bias (e.g., items 2 or 6). The location of the patients was inferred from the GP office location, which may not always reflect the patient’s actual residential setting.

This cross-sectional design prevents the assessment of frailty progression, predictive validity, or clinical outcomes. Important frailty-related variables such as multimorbidity, polypharmacy, cognitive or affective status, and health literacy were not assessed and may have influenced the screening results. CFS assessments, guided by structured definitions, may also be influenced by subjective judgement, particularly in the absence of standardised functional or cognitive data.

Finally, alternative frailty identification methods such as electronic medical record (EMR)-derived indices or phenotype-based tools (for example, Fried frailty criteria) were excluded because of feasibility constraints in routine practice. These alternatives have shown utility in other studies [23,44,53,54] but require further evaluation regarding their practicality and diagnostic performance in primary care.

### 4.5. Implications for Practice and Future Research

The findings of this study offer several practical implications for frailty screening and management in primary care. Most importantly, they support the use of PRISMA-7 with a cut-off ≥ 4 in clinical settings. This threshold reduced overclassification, improved concordance with the CFS, and partially mitigated sex-related bias, particularly the disproportionate identification of men as frail at a cut-off of three. At the same time, it maintained the ability to detect patients with more advanced frailty, as reflected by stable proportions in the group classified as “frail by both tools”.

While a cut-off of three may still be appropriate for use in preventive care or population health settings, where sensitivity is prioritised to capture early-stage or pre-frail individuals, a cut-off of four appears better suited for contexts where resource allocation or care escalation decisions are based on screening results. These findings are consistent with international validation studies that proposed a cut-off of four for improved specificity in relatively healthier populations [39,40,41].

The persistent discrepancy in frailty classification between men and women at a cut-off of three, linked to the scoring of male sex (item 2) in PRISMA-7, highlights the importance of refining the existing tools to promote equity. The removal of this item (i.e., the PRISMA-6 variant) may reduce sex-related misclassification and enhance internal consistency, as previously proposed in the literature [39,40,52]. Further studies are required to evaluate the performance of PRISMA-6 versus PRISMA-7 across subgroups and settings.

IDC emerged as a responsive component of care once frailty was identified, but its activation remained limited and regionally variable. Health systems aiming to implement or scale up frailty screening must address barriers to IDC uptake, especially in urban areas, where care coordination may be more fragmented. To reduce urban–rural disparities in IDC uptake, targeted policy measures may include increased investment in community nursing, improved digital tools for care coordination, and incentive structures for interdisciplinary collaboration in urban districts. Additionally, the observed trend that younger GPs were more likely to activate IDC suggests the potential value of training initiatives to promote integrated care readiness across all age groups and professional cohorts. Investments in community nursing, interprofessional collaboration, and simplified referral processes could help ensure that frailty screening leads to tangible care interventions.

Future research should focus on the following priorities:Validation studies with complete PRISMA-7 and CFS assessments across the full sample will allow for ROC analyses and more robust cut-off calibration.Longitudinal outcome studies examining frailty progression, care responsiveness, hospitalisation, institutionalisation, and mortality by frailty group and screening tools.Equity-focused analyses of tool performance by sex, age, language, or social determinants can inform the refinement of the existing tools.Implementation science approaches to identify the barriers and enablers of IDC activation and explore provider-level factors (e.g., GP engagement and readiness for interprofessional collaboration).Comparative studies of PRISMA-7, PRISMA-6, and alternative instruments, such as the FRAIL scale [49,54] or EMR-derived frailty indices [16,55,56], including their acceptability, diagnostic accuracy, and feasibility in primary care.

Ultimately, systematic frailty screening and stratified care pathways, when designed with attention to tool validity, context, and equity, have the potential to improve patient outcomes and optimise healthcare resource use. As frail older adults represent the highest users of health services [4], investment in appropriate screening strategies is not only clinically but also economically justified.

## 5. Conclusions

This study demonstrates the feasibility and clinical value of systematic, two-step frailty screening in primary care using PRISMA-7 and the CFS. The use of a PRISMA-7 cut-off ≥ 4, rather than the commonly applied cut-off ≥ 3, improved concordance with clinical judgement, reduced sex-related misclassification, and more precisely identified patients with manifest frailty who may benefit from integrated care. The findings support the integration of validated frailty tools into routine general practice, particularly when combined with follow-up assessments and care pathways, such as IDC. However, disparities in frailty classification by sex and geography as well as variations in IDC activation highlight the need for targeted implementation strategies, tool refinement, and provider training. Future research should focus on validating modified frailty screening instruments, assessing clinical outcomes linked to frailty classification and care activation, and evaluating strategies to optimise equitable access to intensified care in vulnerable older adults. Frailty screening holds promise as a cornerstone of personalised and preventive care; however, its benefits depend on accuracy, equity, and actionable follow-up.

## Figures and Tables

**Figure 1 jcm-14-03431-f001:**
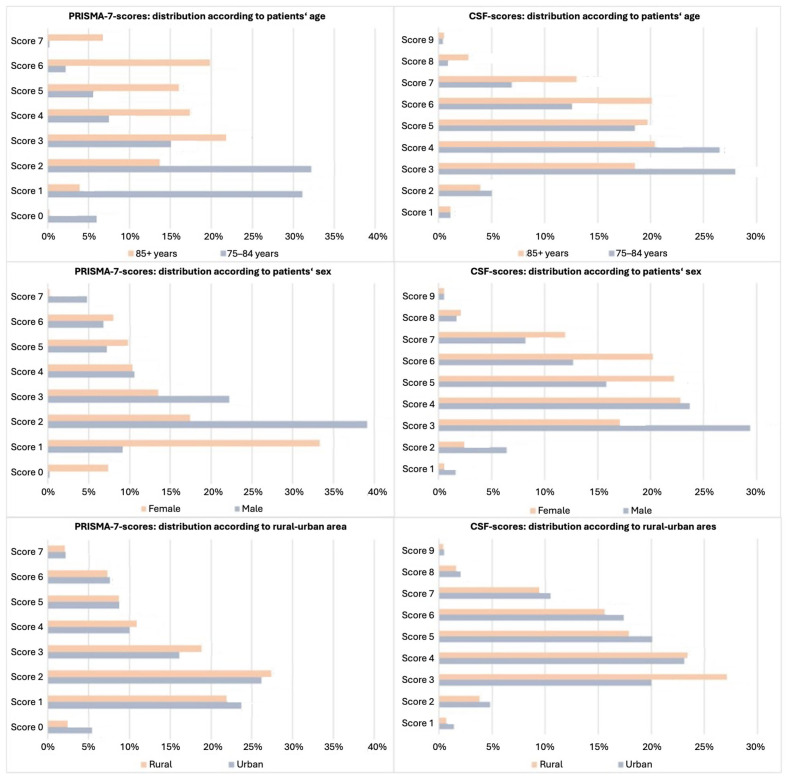
Distribution of the PRISMA-7 and CFS scores by age, sex, and GP office location. Bar charts showing the distribution of PRISMA-7 (**left panels**) and the Clinical Frailty Scale (CFS, **right panels**) scores among patients aged ≥75 years stratified by (**top**) age group (75–84 vs. ≥85 years), (**middle**) sex, and (**bottom**) rural vs. urban location of the general practitioner) office. Most patients aged 75–84 years scored ≤2 on PRISMA-7 and ≤4 on CFS, whereas those aged ≥85 years scored ≥4 on PRISMA-7 and ≥5 on CFS. Female patients showed higher frequencies of CFS scores ≥ 5, while male patients were more likely to score 3 on PRISMA-7. Urban–rural differences in score distributions were modest but suggested slightly higher CFS scores in urban populations. Abbreviations: CFS, Clinical Frailty Scale; GP, general practitioner.

**Figure 2 jcm-14-03431-f002:**
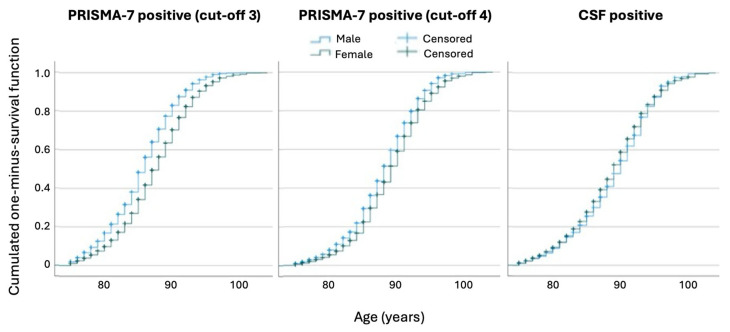
Age of frailty onset by sex, Kaplan–Meier analysis for PRISMA-7 and the Clinical Frailty Scale. Kaplan–Meier curves displaying cumulative frailty prevalence by patient age stratified by sex and frailty assessment tool: PRISMA-7 (cut-offs 3 and 4) and the Clinical Frailty Scale (CFS). With a PRISMA-7 cut-off of 3, men became frail earlier than women (mean age: 85.7 vs. 87.6 years; *p* < 0.001). This difference narrowed at a cut-off of 4 (88.1 vs. 89.2 years; *p* < 0.001). In contrast, CFS showed no significant sex difference in the age of frailty onset (89.2 vs. 88.8 years; *p* = 0.013). These findings highlight how cut-off selection affects sex-specific frailty classification in older adults. Abbreviations: CFS, Clinical Frailty Scale.

**Table 1 jcm-14-03431-t001:** Characteristics of participating general practitioners and screened older adult patients.

Variables	*n* (%)	Median (IQR)
GPs, *n* = 142		
Age, years	–	47 (39–58)
Female sex	72 (50.7)	–
Location of GP office		
Urban area	75 (52.8)	–
Rural area	67 (47.2)	–
Patients, *n* = 19,501		
Age, years	–	81 (78–85)
Female sex	11,203 (60.0)	–
Number of screened patients (PRISMA-7)	18,658	122 (3–173) ^1^

^1^ value per GP.

**Table 2 jcm-14-03431-t002:** Frailty prevalence and score distributions according to PRISMA-7 and the Clinical Frailty Scale (CFS).

Instrument	Measure	Value
PRISMA-7 (cut-off ≥ 3)	Frail patients, *n* (%)	8582 (46.0)
	Median score—frail	4.0 (3.0–5.0)
	Median score—non-frail	1.0 (1.0–2.0)
PRISMA-7 (cut-off ≥ 4)	Frail patients	5372 (28.8)
	Median score—frail	5.0 (4.0–6.0)
	Median score—non-frail	2.0 (1.0–2.0)
CFS	Screened patients	7970
	Frail (score ≥ 5)	3852 (48.3)
	Mild (score 5)	1526 (19.1)
	Moderate (score 6)	1326 (16.6)
	Severe+ (7–9)	1000 (12.5)
	Median score—frail	6.0 (5.0–7.0)
	Median score—non-frail	3.0 (3.0–4.0)

**Table 3 jcm-14-03431-t003:** Distribution of frailty classification according to PRISMA-7 and CFS at the two cut-off levels.

Combined Frailty Status	PRISMA-7 Cut-Off ≥ 3 *n* (%)	PRISMA-7 Cut-Off ≥ 4 *n* (%)
Not frail (PRISMA-7 negative)	10,034 (55.7)	12,910 (71.4)
Frail by PRISMA-7, not frail by CFS	4118 (22.9)	1640 (9.1)
Frail by both PRISMA-7 and CFS	3852 (21.4)	3518 (19.5)

**Table 4 jcm-14-03431-t004:** Predictors of frailty according to PRISMA-7 (cut-offs 3 and 4) and the Clinical Frailty Scale (CFS).

Predictor	PRISMA-7 ≥ 3	PRISMA-7 ≥ 4	CFS ≥ 5
*n*	18,003	17,669	7969
Nagelkerke R^2^	0.327	0.315	0.109
Age (per year)	1.27 (1.26–1.28) ***	1.27 (1.26–1.28) ***	1.09 (1.08–1.10) ***
Female sex	0.54 (0.51–0.58) ***	0.82 (0.76–0.88) ***	1.88 (1.72–2.07) ***
Rural GP office	1.19 (1.11–1.28) ***	n.s.	0.83 (0.76–0.91) ***
Constant	−19.79	−20.45	−7.80

*** *p* < 0.001. Abbreviations: CFS, Clinical Frailty Scale; GP, general practitioner.

**Table 5 jcm-14-03431-t005:** Activation of integrated domiciliary care (IDC) among patients with PRISMA-7 ≥ 3 and CFS ≥ 5 (*n* = 3701). **A.** Activation frequencies. **B.** Descriptive analysis by demographic variables. **C.** Logistic regression models.

**A**				
**Variable**			**Value**	
Eligible patients (PRISMA-7 ≥ 3 and CFS ≥ 5)			3701	
Patients with newly activated IDC			526 (14.2%)	
Median number of IDC activations per GP			3 (IQR 1–5) ^1^	
Minimum–Maximum IDC activations per GP			0–18	
GPs excluded from IDC analysis due to implausible values			3 GPs (*n* = 151 patients)	
**B**				
**Characteristic**	**Group**	**IDC Activated *n* (%)**	**IDC Not Activated *n* (%)**	** *p* ** **-Value ^1^**
Patient age	75–84 years	174 (12.4)	1230 (87.6)	0.013
	≥85 years	352 (15.3)	1945 (84.7)	
Patient sex	Male	209 (14.8)	1201 (85.2)	0.410
	Female	317 (13.8)	1973 (86.2)	
GP office location	Urban	283 (12.4)	1999 (87.6)	<0.001
	Rural	243 (17.1)	1176 (82.9)	
**C**				
**Variable**	**Model with PRISMA-7**		**Model with CFS**	
Nagelkerke’s R^2^	0.076		0.149	
PRISMA-7 score (per point)	OR 1.65 (1.50–1.81), *p* < 0.001		–	
CFS score (per point)	–		OR 2.29 (2.07–2.54), *p* < 0.001	
Rural GP office	OR 1.40 (1.16–1.70), *p* < 0.001		OR 1.47 (1.21–1.79), *p* < 0.001	
GP age (per year)	OR 0.98 (0.97–0.98), *p* < 0.001		OR 0.97 (0.96–0.98), *p* < 0.001	
Patient age/sex	Not significant		Not significant	
GP sex	Not included		Not included	
Constant term	−3.43, *p* < 0.001		−5.75, *p* < 0.001	

^1^ Fisher’s exact test for categorical comparisons. Abbreviations: IDC, integrated domiciliary care; GP, general practitioner.

## Data Availability

The data presented in this study are available upon request from the corresponding author for ethical and privacy reasons.

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
