# Peer review of "Real-World Implementation of PRISMA-7 and Clinical Frailty Scale for Frailty Identification and Integrated Care Activation: A Cross-Sectional Study in Northern Italian Primary Practice"

_jcm, 2025, doi:10.3390/jcm14103431_

Round 1

Reviewer 1 Report

Comments and Suggestions for Authors

This study systematically evaluates the feasibility and clinical applicability of the PRISMA-7 and CFS in frailty screening within primary healthcare practices in northern Italy through large-scale cross-sectional data analysis. The research not only provides detailed data on the performance of the two tools at different thresholds but also reveals the impact of demographic and social factors on frailty classification. These findings provide important evidence for optimizing frailty screening tools, reducing gender and social biases, and promoting the design of personalized care pathways.

However, there are certain limitations in the empirical and policy aspects of the article.

First, in terms of empirical limitations, although the study is large in scale and the data is detailed, there are still some limitations. Firstly, the moderate participation rate and potential selection bias in the doctor sample may affect the generalizability of the results. Secondly, the study failed to conduct a comprehensive accuracy comparison (such as ROC analysis), which limits the in-depth understanding of the tool's performance at different thresholds. Additionally, the low IDC activation rate indicates that there may be implementation barriers between frailty identification and actual intervention, which requires further exploration of the specific reasons.

Second, in terms of policy recommendations, although the study suggests increasing the PRISMA-7 threshold to ≥4 to reduce over classification and gender-related bias, there is a lack of clear strategies to address the differences in IDC activation rates between rural and urban areas. For example, the study did not discuss in detail how to improve care coordination in urban areas or strengthen investment in community healthcare resources. Furthermore, regarding the phenomenon where younger doctors are more likely to initiate IDC, the study did not explore the underlying reasons and potential training needs.

Some points could be added in the discussion to address the potential shortcomings in the empirical aspects. For example, future research should focus on the following areas: (1) a longitudinal study design to assess frailty progression and its impact on hospitalization rates, institutionalization, and mortality; (2) conducting equity analyses across different genders, ages, and social determinants to further optimize screening tools; (3) combining implementation science methods to explore barriers and facilitators of IDC activation, particularly at the provider level (e.g., physician engagement and interprofessional collaboration skills). At the same time, comparing the acceptability, diagnostic accuracy, and feasibility of different frailty screening tools (such as PRISMA-6, FRAIL scale, etc.) will provide more actionable guidance for clinical practice.

Overall, the article has a clear structure, logical thinking, and holds value for frailty screening research. A few improvements would further enhance the paper.

Reviewer 2 Report

Comments and Suggestions for Authors

Dear authors,

Thank you for this extensive and well organized study on practical implementation of frailty screening in routine GP practice and opportunity to read the well thought and structured manuscript. After reading it several times I came to conclusion that do not have major comments, just one suggestion.

The aim is stated in the past tense (line 67) but follow with three specific statements which do not propose action (e.g. compare the two PRISMA-7 thresholds......., identify demographic predictors......, assess the individuals for whom IDC was activated....). 

Author Response

We thank the reviewer for the kind appraisal of the manuscript and for highlighting this stylistic inconsistency. We have revised the aim formulation to align tense and structure.

Inserted text (revised Aim in the Introduction, line ~67):
The aim of this study was to evaluate the real-world application of PRISMA-7 and the Clinical Frailty Scale (CFS) in primary care. Specifically, it aimed to compare the two PRISMA-7 thresholds (cut-off ≥3 vs. ≥4), to identify demographic predictors of frailty classification, and to examine the profile of individuals for whom Integrated Domiciliary Care (IDC) was activated.